

# Characterization of spray-dried Gac aril extract and estimated shelf life of β-carotene and lycopene

Benjawan Thumthanaruk[1,*], Natta Laohakunjit[2,*] and Grady W. Chism[3]

[1] Department of Agro-Industrial, Food and Environmental Technology, Faculty of Applied Science, King Mongkut's University of Technology North Bangkok, Bangkok, Thailand
[2] School of Bioresources and Technology, King Mongkut's University of Technology Thonburi, Bangkok, Thailand
[3] Department of Biology, Indiana University-Purdue University at Indianapolis, Indianapolis, United States of America
[*] These authors contributed equally to this work.

Corresponding author
Benjawan Thumthanaruk,
benjawan.t@sci.kmutnb.ac.th

## ABSTRACT

**Background**. Fresh Gac (*Momordica cochinchinensis*) fruit is rich in carotenoids, mainly β-carotene and lycopene, but these compounds are sensitive to degradation. Spray drying is used to encapsulate the sensitive β-carotene and lycopene with different materials. Only a few reports of using highly branched cyclodextrin (HBCD) have been published. Additionally, studies of β-carotene and lycopene losses in Gac powders during storage are limited. Therefore, the encapsulation of β-carotene and lycopene of Gac aril with HBCD by spray drying at different inlet temperatures were compared. The shelf life of β-carotene and lycopene during storage was also calculated.

**Methods**. The fresh Gac aril was separated and kept frozen before the experiment. Gac aril and water (1:5 w/v) were centrifuged at 8,000 g at 20 °C for 15 min using a high-speed centrifuge (Sorval; Dupont, Wilmington, DE, USA). The supernatant was filtered twice and concentrated until 15° Brix using a rotary evaporator (R-200; Buchi, Flawil, Switzerland). The mixture of concentrated aril extract and highly branched cyclodextrin at 5% (w/v) was dried at three inlet temperatures by a spray dryer (B-290; Buchi, Flawil, Switzerland) with drying air flow rate, compressor air pressure, and feed rate set at 473 L/h, 40 m³/h, and 3 mL/min, respectively . The physicochemical qualities, particle image morphology, and estimated storage time of β-carotene and lycopene were determined.

**Results**. Increased inlet temperatures of spray drying decreased the bulk density, β-carotene, and lycopene content of spray-dried powders significantly. The color values of dried powders had significant differences in yellowness (b⋆) and chroma, but not lightness (L⋆), redness (a⋆), and hue when the inlet temperature increased from 160 °C to 180 °C. The maximum reduction of β-carotene and lycopene observed during storage at 55 °C was 90.88% and 91.11% for 33 and 18 days. For β-carotene, the estimated shelf-life (retention of 50% of β-carotene) was 9.9, 48.4, and 91.6 days at 25 °C, 10 °C, and 4 °C. The shelf-life of lycopene was 26, 176, and 357 days at 25 °C, 10 °C, and 4 °C, respectively. HBCD could be potentially used as an encapsulating agent in spray-dried Gac aril, but the shelf-life of β-carotene and lycopene needs to be improved to be useful as a food ingredient.

## INTRODUCTION

Fresh Gac fruit (*Momordica cochinchinensis* Spreng) can be consumed for aril and pulp parts and is rich in carotenoids, mainly β-carotene and lycopene. Pulp, peel, and aril have significant differences in the quantity of lycopene, beta-carotene, and lutein (*Kubola & Siriamornpun, 2011*). From the whole fruit, Gac aril is the most used and studied source of carotenoids (α-carotene, β-carotene, cis-lycopene, trans-lycopene, lutein, zeaxanthin, cryptoxanthin), vitamin E, vitamin C, and essential fatty acids (omega-3 and omega-6) (*Nhung et al., 2010*; *Kha et al., 2014*; *Kubola, Meeso & Siriamornpun, 2013*; *Oanh et al., 2017*). The quantity of beta-carotene and lycopene in Gac aril is five times higher than in tomatoes (*Vuong et al., 2006*). However, beta-carotene and lycopene are prone to oxidation, thereby reducing their antioxidant activity. For use as a food ingredient, spray drying (*Kha, Nguyen & Roach, 2010*; *Cao-Hoang et al., 2011*; *Chuyen et al., 2019*) and freeze-drying (*Tran et al., 2008*; *Kumkong, Thumthanaruk & Banjongsinsiri, 2015*) have been used to reduce losses of β-carotene and lycopene. Compared to freeze-drying, spray drying is more economically feasible and more widely used for encapsulating β-carotene and lycopene. Different encapsulating materials reported include whey protein concentrate and gum Arabic (*Chuyen et al., 2019*), polylactic acid with and without tween 80 (*Cao-Hoang, Fougére & Waché, 2011*; *Cao-Hoang et al., 2011*), and cyclodextrin (*Polyakov et al., 2004*). With the benefit of cyclodextrin's cyclic structure, the sensitive fat-soluble compounds can be trapped inside the dextrin chain, thereby protecting them from oxidation. One of the commercial drying aids is the highly branched cyclodextrin (HBCD), a highly water-soluble cyclodextrin with a molecular weight of 462 kDa and capable of bonding hydrophobic compounds (*Kaneo et al., 2014*). HBCD has been used for pharmaceutical research to stabilize amphotericin B (*Kaneo et al., 2014*). Spray-dried powder for use in inhalers has been produced using HBCD with different active drugs such as 1-naphthoic acid and rifampicin (*Kadota et al., 2015a*), theophylline and clotrimazole (*Kadota et al., 2015b*), and pyrazinamide and rifampicin (*Tse et al., 2018*). *Kagami et al. (2003)* investigated the oxidative stability of spray-dried fish oil having maltodextrin or HBCD and sodium caseinate as wall materials. Results showed that the oil load level and the type of dextrin strongly affected the inner structure, and the ratio of the oil to dextrin affected the surface of the microcapsule. *Park, Rho & Kim (2019)* compared carnosic acid solubility in rosemary complexed with cycloamylose, branched dextrin, β-cyclodextrin, or maltodextrin. Carnosic acid microcapsules produced with cycloamylose had the highest solubility, antioxidant activity, and antimicrobial activity. HBCD has been applied in spray drying of medical or food applications. However, no report of HBCD has been published in the spray drying of Gac. In addition to the lack of studies using HBCD in spray drying of Gac aril, there is a paucity of relevant information about the shelf life of β-carotene and lycopene in Gac powders.

Therefore, the encapsulation of β-carotene and lycopene of Gac aril with HBCD by spray drying at different inlet temperatures were compared in this study. The shelf-life study and estimated storage time of β-carotene and lycopene were calculated at different storage temperatures.

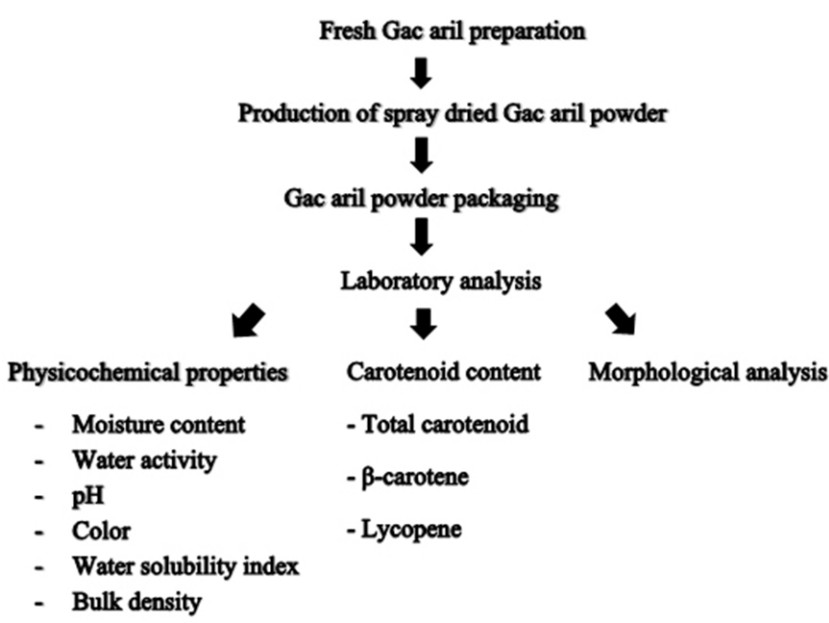

**Figure 1** A schematic diagram of characterization of spray-dried Gac aril powder.

# MATERIALS & METHODS

## Overview of the experimental program

Figure 1 shows a schematic diagram of spray-dried Gac aril powder's production. Briefly, the Gac aril was separated from the whole Gac and kept frozen. The Gac aril was mixed with highly branched cyclodextrin as a drying aid at 5% for spray drying. The condition of inlet temperatures (140 °C, 160 °C, and 180 °C) of spray drying was varied. The spray-dried Gac aril samples were stored at different temperatures (30 °C, 45 °C, and 55 °C) for 45 days. The spray-dried Gac aril samples were then analyzed for physicochemical properties, carotenoid content, and morphological analysis. The results of β-carotene and lycopene were calculated for the spray-dried Gac aril's shelf-life period.

## Purchase of Gac fruit and Custer Dextrin™

Fresh Gac fruit was purchased from a local market in Bangkok, Thailand. Highly branched cyclodextrin (HBCD, Custer Dextrin™) was obtained from a reputable commercial outlet in Bangkok, Thailand.

## Fresh Gac fruit aril preparation

Gac fruit's surface was cleaned with tap water and dried at room temperature. The preparation technique was followed by *Kumkong, Thumthanaruk & Banjongsinsiri (2015)*. The whole fruit was cut and separated from the red aril containing the seeds. The seeds were removed entirely, and the viscous aril pulp was filtered through the cheesecloth. The viscous aril juice was packed in a polypropylene bag and kept frozen at −20 °C until use. The frozen aril was thawed at room temperature for further experiments.

## Production of spray-dried Gac aril powder

The preparation of spray-dried Gac aril was modified the method of *Kha, Nguyen & Roach (2010)*. For each experimental run, the red viscous aril juice (500 g) was blended in distilled water (2.5 L) in a ratio of 1:5. The resulting juice was centrifuged using a high-speed centrifuge (Sorval; Dupont, Wilmington, DE, USA) at 8,000 g at 20 °C for 15 min and filtered twice using a No.1 Watchman filtered paper. Next, the filtered Gac aril juice was then concentrated by a rotary evaporator (R-200; Buchi, Flawil, Switzerland) until the soluble solid was 15° Brix. The concentrated Gac aril juice was mixed with HBCD at 5% (w/v) until the HBCD was completely dissolved. The mixture was dried using a spray dryer (B-290; Buchi, Flawil, Switzerland). Three inlet temperatures (140 °C, 160 °C, and 180 °C) of drying air flow rate, compressor air pressure, and feed rate were set at 473 L/h, 40 m$^3$/h, and 3 mL/min, respectively. After the drying process, the Gac aril powder was collected and packed in a zip-lock bag wrapped with aluminum foil and immediately stored in a desiccator at room temperature. The spray-drying process of Gac aril was carried out in duplicate.

## Analysis
### Moisture content

The moisture content of Gac samples (2 g) was determined by the oven drying method at 105 °C (*AOAC, 2000*).

## Water activity (a$_w$)

The spray-dried powders' water activity was measured using a water activity meter (AquaLab; Decagon Devices, Pullman, W, USA). The water activity meter was calibrated with a saturated salt standard before use. Measurements were performed at room temperature.

### pH

pH was determined by blending 1 g powder with 5 mL of deionized water at room temperature and using a pH meter (Oaklon, USA).

### Color

The color of Gac aril powders was measured in powder and solution. The Gac fruit powder sample was dissolved with deionized water at 1:5 (w/v) and stirred until the powder was completely soluble. The Gac powder or solution was poured into a glass container approximately half cup and measured using a color meter (C-10, Minolta, Japan) calibrated with a standard white tile. The results were expressed as Hunter color values of L*, a*, and b*, where L* was used to denote lightness (0–100), +a* redness and -a* greenness, and +b* yellowness and -b* blueness. Hunter values of the samples for each treatment method were measured in triplicate.

## Water solubility index (WSI)

The water solubility index of the spray-dried Gac powders was analyzed by *Anderson et al. (1969)*. The Gac aril powder (2.5 g) was added to distilled water (30 mL) in a 40 mL centrifuge tube, mixed well, incubated in an incubator at 37 °C for 30 min, and then

centrifuged for 20 min at 10,000 rpm in a high-speed centrifuge (Sorval, Dupont, USA). The supernatant was dried in an oven dryer at $103 \pm 2$ °C. The WSI (%) was calculated as the percentage of dried supernatant with respect to the original 2.5 g Gac aril sample.

## Bulk density

Bulk density was determined by *Goula, Adamopoulos & Kazakis (2004)*. Gac aril powder (2 g) was added into an empty 10 mL graduated cylinder and vortexed on a vortex mixer for 1 min. The bulk density value (g/mL) was calculated by the powder's mass ratio to the volume occupied in the cylinder.

## Determination of total carotenoid content

The total carotenoid content of Gac aril powder was analyzed by the modified method of *Kaisangsri et al. (2016)*. The Gac powder sample of 80 mg and 4 mL of dimethyl sulfoxide (DMSO) was added to the glass tube, incubated at 75 °C in a water bath for 50 min, and cooled to room temperature. After that, the 4 mL n-hexane, containing 0.1 g/100 mL butylated hydroxytoluene, was added, vortexed for 10 s, and left for 30 min. A few (2-3) drops of ethanol were added to precipitate any protein in the hexane phase. The top layer was removed into another tube using a glass pipette. Approximately 0.5 g of anhydrous sodium sulfate was added to the tube and let stand for 15 min to absorb any residual water in the n-hexane solution. Total carotenoid concentration in the hexane solution was then measured absorbance at 450 nm wavelength using a UV–VIS spectrophotometer (UV-1601, Shimadzu, Japan). Total carotenoid concentration was calculated with a standard curve using the known concentrations of β-carotene (0–1 mg/L) in hexane. Triplicate samples were analyzed.

## Determination of β-carotene and lycopene content by HPLC

β-carotene and lycopene contents were measured using *Ishida et al. (2004)* with some modification. Gac aril powder (1 g) was extracted with 5 mL of 40% methyl-tert-butyl ether (MTBE), 50% methanol, and 10% ethyl acetate (v/v) for 1 hr. The extracts were passed through a 0.45-mm poly(tetrafluoroethylene) filter. Throughout these procedures, care was taken to keep samples ice-cold and protect them from light exposure. β-carotene and lycopene content were quantified using a reversed-phase HPLC system, consisting of Agilent 1200 series, Diode-Array Detector, auto-injector, and column temperature regulator. Separations were accomplished using an analytical polymeric Verticep[TM] Bio C30 column (250 × 4.6-mm). The C30 column was then conditioned with elution solvent at a 1 mL/min flow rate for 50 min. Carotenoids were separated using an isocratic mobile phase of 40% methyl-tert-butyl ether (MTBE), 50% methanol, and 10% ethyl acetate (v/v). Injection volumes ranged from 10 µL. The column temperature was maintained at 28 °C. Samples were detected at 450 nm.

## Morphological analysis

Gac aril powders' surface morphology was evaluated using scanning electron microscopy (JSM-7600F, Jeol, Tokyo, Japan). The samples were briefly coated with 5 nm gold under vacuum by a fine auto coater (JSM-7600F, Jeol, Tokyo, Japan), and the powder morphology

was carried out at an accelerating voltage of 1.0 kV by the scanning electron microscopy. The digital images were captured with magnifications of 1000X–5000X.

### Determination of shelf-life of β-carotene and lycopene of spray-dried Gac aril powders

Gac aril powder (10 mg) produced with an inlet temperature of 160 °C was kept in closed vials. The vial samples were then stored at different temperatures of 30 °C, 45 °C, and 55 °C for 45 days. Triplicate samples were periodically withdrawn during the storage to measure β-carotene and lycopene contents by HPLC, as described previously.

β-carotene and lycopene content of storage at 45 °C and 55 °C was used to calculate storage study by accelerated shelf-life tests (ASLT), using the following equation (*ASTM, 1980*; *Mizrahi, 2004*). Temperature coefficient $Q_{10}$ and $Q_1$ are the reaction rate for a different temperature of 10 and 1 °C, respectively, as illustrated in Eqs. (1) and (2). The storage time of β-carotene and lycopene with a 50% reduction at 25 °C, 10 °C, and 4 °C were also calculated by Eq. (3).

$$Q_{10} = \frac{\theta_{s(T)}}{\theta_{s(T+10)}} \tag{1}$$

$$Q_1 = Q_{10}^{0.1} \tag{2}$$

$$Q_1^{\Delta T} = \frac{\theta_{s(T)}}{\theta_{s(T+\Delta T)}} \tag{3}$$

where $\theta_{s(T)}$ is the storage time at the temperature T (days), $\theta_{s(T+10)}$ isthe storage time at the temperature T+10 (days), and ΔT is the difference in °C between the predicted temperature and accelerated temperature.

### Statistical analysis

All analytical measurements were carried out in triplicate. The results were calculated and reported in the mean value and standard deviation. The statistical differences were compared to the mean values using Duncan's multiple range test of SPSS for Windows software version 25.0.

## RESULTS

### Effect of inlet drying temperature on the physicochemical properties of Gac aril juice powder

The spray-dried Gac aril juice powders produced at different inlet temperatures (140 °C, 160 °C, and 180 °C) showed slightly different physicochemical characteristics, as seen in Table 1. The moisture contents ranged from 9.89–10.86%. When the inlet temperature was changed from 140 °C to 160 °C, powders' water activity values decreased from 0.61 to 0.52. No significant change in $a_w$ was obtained when the temperature was increased to 180 °C. The Gac aril solution's pH was in acidic pH (5.53–5.63). For bulk density, the increased inlet temperature significantly reduced the density. The WSI relates to the solubility of dried powder in water. In this study, the aril powder spray dried with an inlet temperature of 160 °C had a different result of WSI compared to those of spray-dried aril powders using inlet temperatures of 140 °C and 180 °C. Spray drying at 140 °C may not be

**Table 1 Characteristics of spray-dried Gac aril powders.**

| Sample/Analysis | Gac aril powder dried at 140 °C | Gac aril powder dried at 160 °C | Gac aril powder dried at 180 °C |
|---|---|---|---|
| Moisture content[ns] (%) | $10.860 \pm 0.630$ | $10.520 \pm 0.480$ | $9.890 \pm 0.550$ |
| $a_w$ content | $0.610^a \pm 0.002$ | $0.520^b \pm 0.002$ | $0.520^b \pm 0.002$ |
| pH | $5.630^a \pm 0.008$ | $5.540^b \pm 0.017$ | $5.530^b \pm 0.016$ |
| Bulk density (g/mL) | $0.630^a \pm 0.006$ | $0.580^b \pm 0.004$ | $0.570^c \pm 0.003$ |
| Water solubility index (%) | $81.200^b \pm 1.316$ | $85.500^a \pm 2.071$ | $81.660^b \pm 1.020$ |
| Total carotenoid compound (mg $\beta$-carotene/g) | $0.819^a \pm 0.045$ | $0.716^b \pm 0.043$ | $0.640^b \pm 0.050$ |
| $\beta$-carotene (mg/g) | $1.403^a \pm 0.019$ | $1.217^b \pm 0.017$ | $1.071^c \pm 0.007$ |
| Lycopene (mg/g) | $0.225^a \pm 0.004$ | $0.141^b \pm 0.018$ | $0.102^b \pm 0.018$ |

**Notes.**
Values are expressed as mean$\pm$standard deviation ($n = 3$).
Numbers with different small superscripts in the same column showed significant differences ($p < 0.05$).
ns refers to no significant differences

**Table 2 Color of spray-dried Gac aril powders.** Values are expressed as mean$\pm$standard deviation ($n = 3$) Numbers with different small superscripts in the same columns showed significant differences ($p < 0.05$) ns refers to no significant differences.

| Sample | L* | a* | b | Chroma | hue |
|---|---|---|---|---|---|
| Gac dried powder at 140 °C | $84.91^a \pm 0.61$ | $10.84^c \pm 0.18$ | $24.23^c \pm 0.40$ | $26.55^c \pm 0.43$ | $65.90^b \pm 0.03$ |
| Gac dried powder at 160 °C | $81.95^b \pm 0.29$ | $12.66^b \pm 0.07$ | $27.40^b \pm 0.15$ | $30.19^b \pm 0.17$ | $65.20^c \pm 0.01$ |
| Gac dried powder at 180 °C | $80.49^c \pm 0.26$ | $14.02^a \pm 0.07$ | $32.01^a \pm 0.23$ | $34.94^a \pm 0.23$ | $66.35^a \pm 0.04$ |

**Notes.**
Values are expressed as mean$\pm$standard deviation ($n = 3$).
Numbers with different small superscripts in the same columns showed significant differences ($p < 0.05$).
ns refers to no significant differences.

microbiologically stable as the $a_w$ was higher than 0.60 and might cause aggregation during storage.

Total carotenoid compounds (TCC) of β-carotene and lycopene significantly decreased at increased inlet drying temperature (Table 1). Increased inlet temperatures showed significantly affected color parameters (Table 2). When reconstituted in water, the solution of Gac aril yielded no significant differences in the lightness (L*), redness (a*), and hue (Table 3). The yellowness (b*) and chroma values increased when the inlet temperature increased from 160 °C to 180 °C, but not from 140 °C to 160 °C. The decreased L* and increased b* could be affected by high drying temperature.

## Morphology of spray-dried Gac aril powders by SEM

The morphology of the dried particles showed the cluster of spherical particles with a mixed size of greater than 10 μm when the increased temperature was used (Fig. 2). The scar remaining on the particle resulted from the detached particle. No broken particles were seen in all spray-dried samples.

**Table 3 Color of reconstituted Gac aril solutions.** Values are expressed as meanstandard deviation ($n = 3$) Numbers with different small superscripts in the same column showed significant differences ($p < 0.05$) ns refers to no significant differences.

| Sample | L*ns | a*ns | b | Chroma | huens |
|---|---|---|---|---|---|
| Gac dried powder at 140 °C and solubilized in water | 52.35 ± 0.10 | 10.78 ± 0.26 | 36.76[b] ± 0.39 | 38.31[b] ± 0.42 | 73.66 ± 0.29 |
| Gac dried powder at 160 °C and solubilized in water | 52.08 ± 0.07 | 11.36 ± 0.28 | 37.25[b] ± 0.79 | 38.95[b] ± 0.80 | 73.04 ± 0.34 |
| Gac dried powder at 180 °C and solubilized in water | 52.22 ± 0.30 | 11.78 ± 0.59 | 40.05[a] ± 1.44 | 41.74[a] ± 1.55 | 73.62 ± 0.24 |

**Notes.**
Values are expressed as meanstandard deviation ($n = 3$).
Numbers with different small superscripts in the same column showed significant differences ($p < 0.05$).
ns refers to no significant differences.

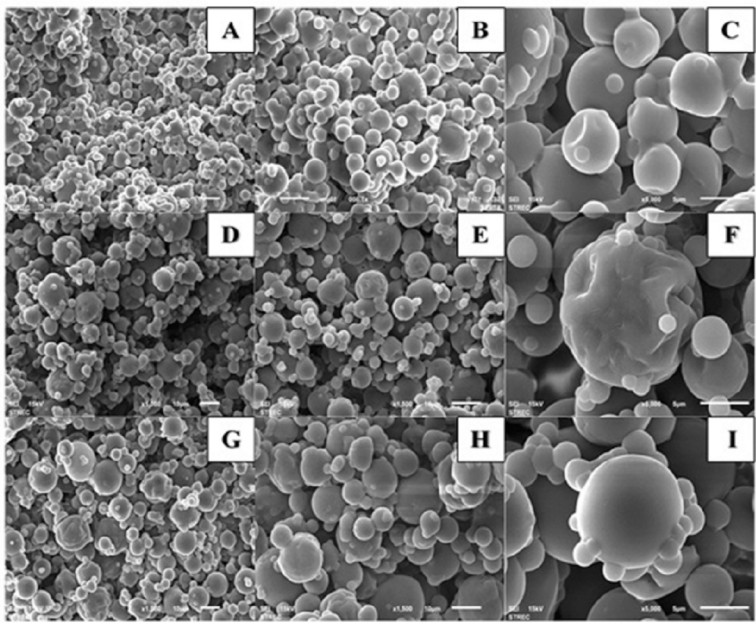

**Figure 2 Scanning Electron Microscopic (SEM) images of Gac aril powders dried at different spray drying inlet temperatures; A–C: at 140 °C; D–F: at 160 °C; G–I: at 180 °C Magnification: A, D, G × 1,000; B, F, H × 1,500; C, F, I × 5,000.**

## Effect of storage temperature and time on the reduction of β-carotene and lycopene in spray-dried Gac aril powder

The concentration of β-carotene and lycopene content of spray-dried Gac aril powders reduced as affected by the increasing storage temperature and time. At 30 °C, β-carotene and lycopene were reduced from 1.403 ± 0.019 and 0.225 ± 0.004 to 0.117 ± 0.014 and 0.033 ± 0.004 mg/g in samples kept for 42 and 24 days, respectively (File 1). Increased storage temperature shortened the shelf life of β-carotene and lycopene. The β-carotene reductions were 90.887 and 89.950% at 33 and 21 days, and the lycopene reductions were 65.778 and 91.111% at 21 and 18 days when kept at 45 °C and 55 °C, respectively (Figs. 3A–3B and Figs. 4A–4B). The reduction of β-carotene and lycopene was approximately 12% and 10%

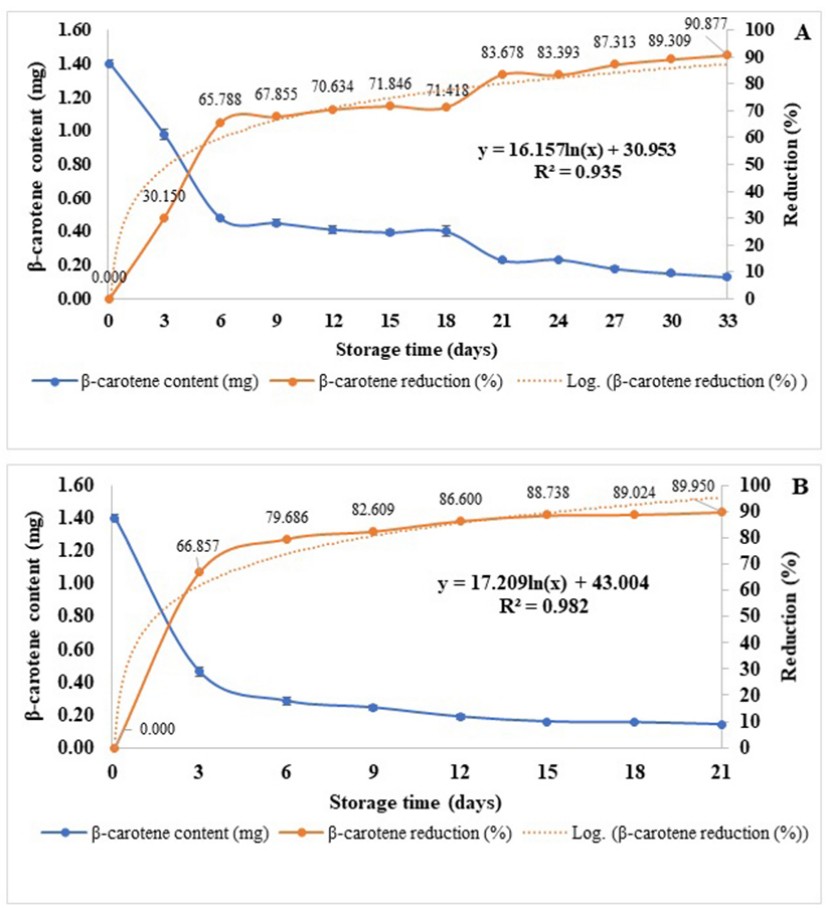

**Figure 3** **$\beta$-carotene content and its reduction (%) of spray-dried Gac aril powder stored at 45 °C (A) and 55 °C (B).**

between 18–21 days of storage, respectively (Figs. 3A and 4A). The $Q_{10}$ for the reduction of $\beta$-carotene and lycopene were 1.112 and 1.135 (Table 4).

The estimation of $\beta$-carotene and lycopene storage times if kept at 25 °C, 10 °C, and 4 °C is in Table 5. The regression equations from Figs. 3A–3B and Figs. 4A–4B, assuming a 50% loss of $\beta$-carotene or lycopene, were used for this calculation. Results showed that if the spray-dried aril powders were kept at 4 °C, the shelf-life of $\beta$-carotene and lycopene were 92 and 357 days (Table 5). Therefore, the retention of $\beta$-carotene and lycopene showed a pronounced effect when encapsulated with the Custer Dextrin$^{\text{TM}}$.

## DISCUSSION

Spray drying has been used to encapsulate Gac's carotenoid compounds (*Chuyen et al., 2019*; *Cao-Hoang, Fougére & Waché, 2011*; *Cao-Hoang et al., 2011*; *Kha, Nguyen & Roach, 2010*; *Tran et al., 2008*). Several factors affecting the degradation of $\beta$-carotene and lycopene, such as the type and concentration of wall materials and inlet temperatures of spray drying, have been studied. The wall material plays a vital role in protecting sensitive

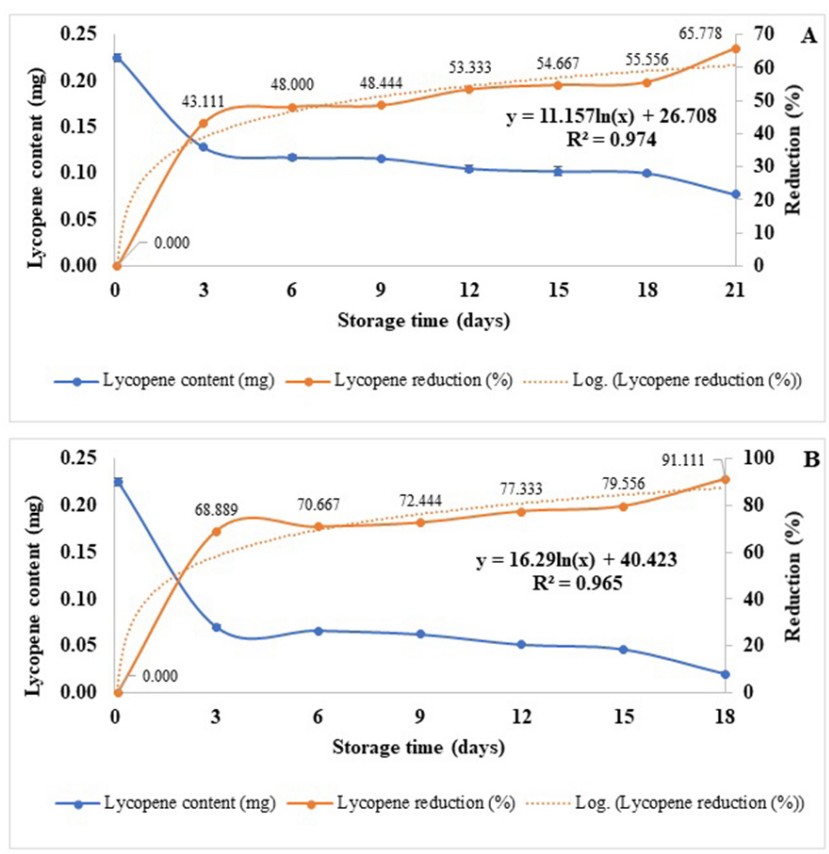

**Figure 4  Lycopene content and its reduction (%) of spray-dried Gac aril powder stored at 45 °C (A) and 55 °C (B).**

carotenoid compounds. Different types of wall material used were a mixture of whey protein concentrate and gum Arabic (*Chuyen et al., 2019*) and polylactic acid (*Cao-Hoang, Fougére & Waché, 2011*; *Cao-Hoang et al., 2011*), and maltodextrin (*Kha et al., 2014*; *Kha, Nguyen & Roach, 2010*). The HBCD was first reported to encapsulate carotenoid-rich oil lycopene at the lowest concentration of 5%. With differences in experimental factors studied, the expected results of spray-dried Gac powders were also different. Results of moisture content and $a_w$ (Table 1) were different from *Chuyen et al. (2019)* that the spray-dried carotenoid-rich oil from Gac peel with a mixture of whey protein concentrate and Arabic gum powder had the moisture content of 3.0–5.2% and $a_w$ of 0.32–0.45 when applied inlet temperature of 160 °C and 180 °C, respectively. The characteristics of spray-dried Gac aril with 5% HBCD were mostly lower than the results of *Kha, Nguyen & Roach (2010)* that applied inlet temperatures from 120 °C to 200 °C and maltodextrin 10, 20, and 30% to Gac aril, except for the moisture content and WSI. The spray-dried Gac had moisture content, $a_w$, pH, bulk density, and WSI ranged from 3.88–5.29%, 0.45–0.50, 4.12–4.45, 0.66–0.78 g/ml, and 37.13–37.62% (*Kha, Nguyen & Roach, 2010*).

**Table 4 Shelf life calculation of $\beta$-carotene and lycopene at 45 °C and 55 °C.**

| Compound | Shelf life calculation | Equation | Results |
|---|---|---|---|
| $\beta$-carotene | Storage at 45 °C | $y = 16.157 \ln(x) + 30.953$ | $x = 1.179$ days |
| | Storage at 55 °C | $y = 17.209 \ln(x) + 43.004$ | $x = 0.407$ days |
| | $Q_{10}$ | $= \dfrac{\theta_{s(45)}}{\theta_{s(55)}}$ | $= \dfrac{1.179}{0.407} = 2.897$ |
| | $Q_1$ | $= Q_{10}^{0.1}$ | $= 2.897^{0.1} = 1.112$ |
| Lycopene | Storage at 45 °C | $y = 11.157\ln(X) + 26.708$ | $X = 2.008$ days |
| | Storage at 55 °C | $y = 16.290\ln(X) + 40.423$ | $X = 0.590$ days |
| | $Q_{10}$ | $= \dfrac{\theta_{s(45)}}{\theta_{s(55)}}$ | $= \dfrac{2.008}{0.590} = 3.539$ |
| | $Q_1$ | $= Q_{10}^{0.1}$ | $= 3.539^{0.1} = 1.135$ |

**Table 5 Estimation of shelf life of $\beta$-carotene and lycopene at 25 °C, 10 °C, and 4 °C.**

| Criteria | Shelf-life calculation | Equation | Result ($\theta_s$) (day) |
|---|---|---|---|
| With 50% reduction of $\beta$-carotene | at 25 °C | $Q_1^{45-25} = \dfrac{\theta_{s(25)}}{\theta_{s(45)}}$ | $1.112^{20} = \dfrac{\theta_{s(25)}}{1.179}$ $\theta_{s(25)} = 9.854$ days |
| | at 10 °C | $Q_1^{45-10} = \dfrac{\theta_{s(10)}}{\theta_{s(45)}}$ | $1.112^{35} = \dfrac{\theta_{s(10)}}{1.179}$ $\theta_{s(10)} = 48.437$ days |
| | at 4 °C | $Q_1^{45-4} = \dfrac{\theta_{s(4)}}{\theta_{s(45)}}$ | $1.112^{41} = \dfrac{\theta_{s(4)}}{1.179}$ $\theta_{s(4)} = 91.582$ days |
| With 50% reduction of lycopene | at 25 °C | $Q_1^{45-25} = \dfrac{\theta_{s(25)}}{\theta_{s(45)}}$ | $1.135^{20} = \dfrac{\theta_{s(25)}}{2.088}$ $\theta_{s(25)} = 26.281$ days |
| | at 10 °C | $Q_1^{45-10} = \dfrac{\theta_{s(10)}}{\theta_{s(45)}}$ | $1.135^{35} = \dfrac{\theta_{s(10)}}{2.088}$ $\theta_{s(10)} = 175.625$ days |
| | at 4 °C | $Q_1^{45-4} = \dfrac{\theta_{s(4)}}{\theta_{s(45)}}$ | $1.135^{41} = \dfrac{\theta_{s(4)}}{2.088}$ $\theta_{s(4)} = 357.458$ days |

The stability of color of Gac's β-carotene and lycopene having the red-orange shade has become of interest for replacing the synthetic color for food products (*Selig et al., 2018*; *Kumkong et al., 2020*); therefore, the protection of these sensitive compounds from processing has become necessary. The color values of encapsulated spray-dried Gac with HBCD were similar to the spray-dried Gac with maltodextrin if the inlet temperatures were not greater than 180 °C (*Kha, Nguyen & Roach, 2010*). However, with different drying, the spray-dried Gac aril's color yielded a higher chroma value than that of freeze-dried Gac aril (*Kumkong et al., 2020*).

The differences in drying agents and inlet temperatures influence the reduction of carotenoid content of spray-dried Gac aril. This study showed total carotenoid content of 0.819, 0.716, and 0.640 mg β-carotene/g when used inlet temperature of 140 °C, 160 °C, and 180 °C (Table 1). The values were lower than the study of *Kha, Nguyen & Roach (2010)* that showed the total carotenoid content of 1.95 mg/g when used at the inlet temperature of 120 °C and maltodextrin 10% (w/v). *Tran et al. (2008)* reported that the spray-dried Gac aril had carotenoid content of 379.7 µg/g powder with almost no β-carotene when used the inlet temperature 200 °C. The study confirmed that the reduction of β-carotene and

lycopene occurred during the spray drying. In this study, the concentration of HBCD was used only at 5%. However, it is worth to study further the effect of increased concentration of HBCD on the retention of Gac aril carotenoid.

The scanning electron microscope can be used to observe the microencapsulated of Gac aril carotenoid. The use of HBCD encapsulated Gac carotenoid displayed mostly round and smooth surface particles (Fig. 2). Compared to *Chuyen et al. (2019)*, which displayed the SEM pictures of whey protein concentrated and Arabic gum wall material encapsulated with Gac carotenoids, the shape and size of particles were quite different. The spray-dried powder with different wall materials influences particles' integrity and affects total carotenoid content loss during drying (*Chuyen et al., 2019*; *Kha, Nguyen & Roach, 2010*).

The β-carotene and lycopene are sensitive carotenoid compounds that can be degraded by pretreatment (*Tran et al., 2008*; *Kha et al., 2014*), light (*Nhung et al., 2010*), high temperature of the process (*Tran et al., 2008*; *Kha, Nguyen & Roach, 2010*), oxygen (*Nhung et al., 2010*; *Cao-Hoang, Fougére & Waché, 2011*) and storage time (*Nhung et al., 2010*; *Chuyen et al., 2019*). Results showed that the shelf-life of β-carotene and lycopene was affected by increased temperature to 10 °C and 25 °C (Table 5). From the calculation, the degradation of β-carotene showed more pronounced than that of lycopene shown in Figs. 3A–3B and Figs. 4A–4B. The retention of β-carotene of this study differs from the study of *Chuyen et al. (2019)*, who reported the total carotenoid retention of the encapsulated powder of Gac peel oil stored at 5 °C and 20 °C for 6 months was 65.3% and 41.5%, respectively, despite differences in wall material and spray-drying condition. The structure of these two compounds is different. β-carotene consists of isoprene units having beta-rings at the ends. Once the β-carotene molecule is hydrolyzed, the two retinal molecules designated as vitamin A are formed, and lycopene is a polar acyclic carotenoid. These two compounds can be undergone oxidations. *Cao-Hoang et al. (2011)* illustrated the chemical degradation pathway of both β-carotene to form compounds of β-ionone, 5,6-epoxy- β-ionone, dihydro actinidiolide, and β-cyclocitral. The shelf-life of β-carotene and lycopene in this study were different and not comparable from other research works due to several factors, such as different drying agents used, spray drying conditions, storage condition, and analysis method. *Cao-Hoang et al. (2011)* stated that the lycopene was more sensitive to oxidation than β-carotene. The degradation rate of lycopene prepared with PLA and without PLA was 11 mM and 7 mM after 24 h of oxidation. *Chuyen et al. (2019)* reported the low retention of total carotenoids in spray-dried Gac oil powder encapsulated with whey protein concentrate and Arabic gum was 65.3% and 41.5% after kept for 6 months at 5 °C and 20 °C.

## CONCLUSIONS

The spray-dried Gac aril powders with 5% HBCD produced with different inlet temperatures (140 °C, 160 °C, and 180 °C) showed slightly different physicochemical characteristics (moisture content, $a_w$, bulk density, and WSI). Drying with HBCD enclosed the carotenoids in the spherical particles with a mixed size greater than 10 μm. The

use of HBCD and spray drying at the inlet temperature of 160 °C showed potential for encapsulating the sensitive compounds of β-carotene and lycopene. The shelf-life study confirmed that the low-temperature storage at 4 °C of the spray-dried aril powders with 50% reduction β-carotene and lycopene content was 91 and 357 days, respectively.

## ACKNOWLEDGEMENTS

We would like to thank Mr. Keerati Thamapan for performing statistical analysis and calculation.

### Funding

This research was funded by King Mongkut's University of Technology North Bangkok under contract no. KMUTNB-60-GOV-2560A11902083. The funders had no role in study design, data collection and analysis, decision to publish, or preparation of the manuscript.

### Grant Disclosures

The following grant information was disclosed by the authors:
King Mongkut's University of Technology North Bangkok: KMUTNB-60-GOV-2560A11902083.

### Competing Interests

The authors declare there are no competing interests.

### Author Contributions

- Benjawan Thumthanaruk conceived and designed the experiments, performed the experiments, analyzed the data, prepared figures and/or tables, authored or reviewed drafts of the paper, and approved the final draft.
- Natta Laohakunjit conceived and designed the experiments, performed the experiments, analyzed the data, prepared figures and/or tables, and approved the final draft.
- Grady W. Chism analyzed the data, authored or reviewed drafts of the paper, and approved the final draft.

### Data Availability

Raw data are available in the Supplemental Files.

### Supplemental Information

Supplemental information for this article can be found online at http://dx.doi.org/10.7717/peerj.11134#supplemental-information.

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
