# Peer review of "Characterization of spray-dried Gac aril extract and estimated shelf life of β-carotene and lycopene"

_PeerJ, doi:10.7717/peerj.11134_

## Round 0.1 · original submission · Major Revisions

Thank you authors for submitting your manuscript for consideration at PeerJ. Reviewers have considered your work and found a lot of merits. They also had some queries, which deserve your attention, to improve the scientific standing of the work.

Please, kindly attend to the queries, one by one. In addition, kindly consider the following:

a) Title: Kindly modify it to 'Characterization of spray-dried Gac aril powder and estimated shelf life of β-carotene and lycopene at different inlet temperatures'

b) Abstract: Why accelerated study? Kindly rethink through well about this terminology

c) Introduction : Kindly divide the introduction into two clear paragraphs, first paragraph should end with sentence ' The Cluster Dextrin™ has been used for pharmaceutical research with the benefit of stabilizing and encapsulating of amphotericin B (Kaneo et al., 2014). '

Second paragraph should start with sentence:
There appears to no study that has investigated Cluster Dextrin™ employed in the spray drying of Gac aril. Continue this paragraph by adding literature about studies that have applied Cluster Dextrin™ in spray drying, and what their aims were. Synthesize some literature here, to justify the use of Cluster Dextrin™. Then, you can place this sentence 'In addition to lack of studies about Cluster Dextrin™ employed in the spray drying of Gac aril, there is paucity of relevant information concerning.....Gac β-carotene and lycopene. '
By the way, please what do you actually mean by accelerated study? Does this reflect the actual parameters studied? You characterized the spray-dried Gac aril powder, and estimated shelf life, isn't it? Please kindly stick to these words.

d) Materials and methods: Kindly start this section with a subheading 'overview of the experimental program'. This is necessary, to provide readers a snapshot of the entire study. There should be a schematic diagram: collection of samples >Fresh Gac fruit aril preparation>Production of spray-dried Gac aril powder>Gac aril powder packaging>Laboratory analysis > (indicate all the analysis, showing the total number of samples allocated to each)
Please, kindly ensure that permission has been given by Glico Nutrition Co., Ltd. to indicate/use their name. If not, I suggest you revise it ( for example: The commercial Custer 79Dextrin™ was obtained from a commercial reputable outlet in Bangkok, Thailand, and permission was not given to use their name.)

e) Results and Discussion: Please, separate the results and discussion, as per PeerJ standard rules. The discussion appears ok. It would be useful to ensure that appropriate and sufficient relevant synthesized literature is used in the discussion. Authors, please, apply your discretion to do this.

The authors have performed a brilliant study. Looking forward to your revised manuscript. Thank you very much for considering PeerJ as your journal of choice, and we look forward to more of your scholarly contributions.

Reviewer 1 ·

Basic reporting

well described

Experimental design

well described

Validity of the findings

well described

Additional comments

the manuscript is well written and obtained good results. Congratulation!

Introduction: well described

Materials and method:
- Line 90: Rpm to g, also abstract
- Line 96-97: there are 3 constant variables, drying air flow rate, compressor air pressure and feed rate. But only 2 numbers are shown.
- Line 145: it would be better if word “wavelength” is added after “450 nm”.

Results and discussion: well described
- There is an interesting pattern from figure 2A in the B-carotene reduction and content between 18-21 days which might be important to be explained

Table and figure:
- The data should be better if are shown in 3 decimal
- Consistency in writing ß in figure 2
- The statistical analysis of the differences in carotene and lycopene content during the storage is important to be shown (whether its in a different table or adding the error bar in figure 2).

·

Basic reporting

The manuscript presents one of the interesting properties of gac fruit. However, the clarity and sentence structure of the manuscript require improvement. For example, the second sentence in the introduction requires rephrasing for more clarity.

Experimental design

The authors should provide values for the parameters analyzed before initial drying. This should be included in Table 1. Alternatively, they can explain why the data were not included in the report (or the experimental design).

Validity of the findings

1. The values of the β-carotene found in this study are lower (Kindly check https://doi.org/10.1016/j.anres.2016.04.003). The authors should explain the possible scientific-based reasons for their results.

2. Make more comparison of the values obtained in your study with those obtained by other researchers for similar studies

3. Improve your sentence structure and grammatical correctness. For example, in line 252 and 253, where you stated that "...β-carotene and lycopene reduced from 1.403±0.019 and 0.225±0.004 to 0.117±0.014 and 0.033±0.004 mg/g sample after kept for 42 and 24 days (data not shown)", you need to add the word "respectively" after the sentence; except if stated otherwise.

Additional comments

Major revision is required as highlighted

---

## Round 0.2 · Minor Revisions

Authors, please, although one reviewer has indicated acceptance, the other has provided comments, hence, minor revision. Kindly address the comments raised.

The editor also encourages authors to:

1) The overview of experimental program section, kindly add more information to this section. Do not just state what the figure shows. Tell the reader more about the figure, in the context of the work. In the end, inform the reader how this figure is linked to the research question of this study. Reiterate the specific objective of this study, so that the reader will see that this schematic flow, represents the entire study.

2) In the next section, 'Materials', change this subtitle to 'Purchase of Gac fruit, and Custer Dextrin™ ', and please make the font size be the same as that of other subsection captions/titles.
Look forward to the revision. Thank you in advance for your great efforts.

Reviewer 1 ·

Basic reporting

It has been improved

Experimental design

Well described

Validity of the findings

well described

Additional comments

General comments:
All comments from the previous review have been addressed well, thank you. However, there are some minor comments:
RESULTS:
- Line 221-222: it is stated “This study showed no effect by spray-drying temperature on the water solubility of spray-dried aril powder”. In Table 1, water solubility index shows temperature 160 C had a higher WSI which is shown by statistical difference.Please check and confirm.
- Line 229-230: based on Table 2, all colour parameters are significantly affected. But it is written “....except for lightness”. Please check and confirm.
- Line 233-234: “the decreased in L…”from table 2, there is no difference in L, please confirm what this sentence means.
Line 235: delete (.)

DISCUSSION
- Line 272-273: delete “and, in this study, the Custer DextrinTM.”
- Line 284: add “respectively” at the end of the sentence.
- Line 292, it is mentioned that differences in drying agents influence the reduction of carotenoid. But, in - line 299, it is stated that this study confirmed that the reduction occurred regardless the type of the wall material used. Does this mean that the result obtained in contrast with the other papers? If so, please state clearly. Or, if not, please make it more clear.

·

Basic reporting

Satisfactory

Experimental design

Satisfactory

Validity of the findings

Satisfactory revisions

Additional comments

Satisfactory revisions

---

## Round 0.3 · Minor Revisions

The authors have satisfactorily addressed comments raised, and have significantly improved the manuscript. However, before accepting the manuscript for publication, the editor requests authors to have a fluent English speaker (or copy-editing service) go through the entire manuscript, to further elevate/improve the standard for readership. Looking forward to receiving the revised manuscript, for acceptance.

Reviewer 1 ·

Basic reporting

Satisfy

Experimental design

Satisfy

Validity of the findings

Satisfy

Additional comments

Thank you very much for your revisions and congratulations!

---

## Round 0.4 · accepted · Accept

The revised manuscript is acceptable for publication. Thank you for your scholarly contribution, and for finding PeerJ as your journal of choice. The editor believes that authors benefitted greatly from the peer-review process. PeerJ looks forward to receiving your future scholarly contributions. Thank you very much.